# Spatio-Temporal Evolution and Driving Factors of Landscape Pattern in a Typical Hilly Area in Southern China: A Case Study of Yujiang District, Jiangxi Province

**Jiajia Zhang** [1,2], **Xiaomin Zhao** [1,2,*], **Jiaxin Guo** [1,2], **Yanru Zhao** [1,2], **Xinyi Huang** [1,2] **and Miao Long** [1,2]

1   Academy of Land Resource and Environment, Jiangxi Agricultural University, Nanchang 330045, China
2   Key Laboratory of Agricultural Resources and Ecology, Poyang Lake Watershed of Ministry of Agriculture and Rural Affairs in China, Jiangxi Agricultural University, Nanchang 330045, China
*   Correspondence: zhaoxm889@126.com

**Abstract:** As the most intuitive manifestation of land use/land cover change, the spatio-temporal evolution of landscape patterns has significant implications for optimizing regional landscape pattern and land use management. Based on multi-period remote sensing data, we selected an optimal scale (570 m) and used the geographic detector model to analyze the spatio-temporal changes in the landscape pattern of a typical hilly area (Yujiang District, Yingtan City, Jiangxi Province) in southern China. The results showed that from 2009 to 2018, the area of urban land, other construction land, rural residential land, and cultivated land expanded by 33.27%, 21.23%, 19.42%, and 1.07%, respectively. In contrast, the area of grassland, forest land, and water area shrank by 18.18%, 5.41%, and 2.19%, respectively, over the past 10 years. At the landscape level, the patch shape became more complex over time, with increased landscape fragmentation and diversity. At the class level, cultivated land, forest land, and grassland tended to be fragmented, whereas rural residential land exhibited an aggregation tendency. Slope gradient, gross regional product, and distance from major highways had a strong ability to explain the spatial differences in landscape pattern change. The results of this study enable a dynamic understanding of landscape pattern evolution in typical hilly areas in southern China and provide a reference for landscape pattern optimization in similar geomorphic settings.

**Keywords:** landscape pattern; spatio-temporal evolution; geodetector; Yujiang District

## 1. Introduction

Human land use patterns across spatio-temporal scales influence land use/cover change, which is most directly reflected in the spatio-temporal evolution of landscape patterns [1]. A landscape pattern is the arrangement of different types and quantities of landscape elements in spatial structure and location. It is the ultimate manifestation of the combined action of natural and socio-economic factors over complex spatio-temporal scales [2,3]. Landscape metrics can highly generalize and explain landscape pattern change, providing an important tool to study landscape pattern evolution [4,5].

The rapid development of remote sensing technology and geographic information systems has enabled research into regional landscape pattern evolution using landscape metrics based on multi-period land use remote sensing image data, which has become a popular topic of land use change in landscape ecology. A growing number of studies have been conducted, including the analysis of land use dynamic evolution, the identification and evaluation of landscape ecological risk areas, and the prediction of future landscape pattern development. The previous studies are mainly concentrated in rapidly urbanized areas and eco-environmentally sensitive and vulnerable areas [6–8].

In China, Ma et al. [9] investigated the dynamic evolution of the landscape pattern of the Yangtze River Economic Belt and its eco-environmental effects. They found a spatial

dislocation between regional natural landscape and artificial landscape in terms of the development level. In contrast with the artificial landscape, the natural landscape showed decreased aggregation and increased fragmentation and diversity. Additionally, Zuo et al. [10] characterized the spatio-temporal variation in landscape ecological risk based on an optimal scale in the southwest mountainous area of Hubei Province, where the Three Gorges Hydraulic Project is located. An overall downward trend was observed for landscape ecological risk in the study area, with a staggered distribution of localized high risk and low risk.

Furthermore, Calvo-Iglesias, M.S.et al. [11] described the change in farmland landscape patterns across northwestern Spain over the past 40 years based on landscape metrics analysis, and landscape fragmentation was identified to be the most important factor leading to the transformation of different types of landscape patches. Despite the myriad of studies on regional integrated landscape patterns, less is known about the overall landscape pattern and variation in class-level landscape metrics at the county scale.

Unraveling the key factors driving landscape pattern evolution is necessary to ascertain its underlying mechanisms, which has significant implications for landscape layout optimization and rational land use arrangement [12]. Kefalas et al. [13] identified distinct factors driving land use cover change in natural vegetation areas and artificial planting areas on Mediterranean islands. Ma et al. [14] characterized periodic landscape pattern changes in the Shule River Basin in northwestern China over the past 30 years. Due to subjective and objective reasons, it is difficult to collect a full set of the natural and socio-economic factors that drive landscape pattern change.

Previous research has explored the mechanisms driving landscape pattern evolution mainly using traditional models based on qualitative analysis, correlation analysis, and regression analysis, which have limitations such as strong subjectivity, strict application conditions, and poor scale suitability [15–18]. Compared with the traditional models, the geographical detector model (geodetector) can more accurately identify the mechanisms driving landscape pattern evolution, given its ability to detect spatial differentiation. It determines the relative influence of an independent variable on the dependent variable based on the similarity in their spatial distribution, and as such, identifies the key factor driving the spatial differentiation in the dependent variable [19,20].

The Yujiang District, located in northeastern Jiangxi Province, is a transition zone from the Wuyi Mountains to the Poyang Lake plain in China. As a typical hilly county, Yujiang District is part of an important passage connecting the mainland to the southeastern coastal area, so its ecological function cannot be ignored. Based on multi-period remote sensing data, we selected the optimal scale for landscape analysis of Yujiang District to explore the spatio-temporal changes in the overall landscape pattern and different landscape classes in terms of landscape fragmentation, patch shape, and diversity. We additionally adopted the geodetector to identify the major factors driving landscape pattern evolution in the study area. Findings of this research advance the understanding of landscape pattern dynamics in typical hilly areas and contribute to sustainable landscape planning and management as well as effective use and ecological protection of land resources in southern China.

## 2. Materials and Methods

### 2.1. Study Area

Yujiang District (28°04′–28°37′ N, 116°41′–117°09′ E) is located in the middle and lower reaches of the Xinjiang River, in Yingtan City, Jiangxi Province, China. It is a narrow land area that extends from south to north. The terrain is generally high in the north and south but low in the middle, and it can be divided into the northern hills, central valleys, and southern hills (Figure 1). The major land use types are cultivated land and forest land. This area has a subtropical monsoon climate with annual averages of 17.6 °C and 1788 mm in terms of temperature and rainfall, respectively. By the year 2018, the population had reached 399,900 and the gross regional product was 13.983 billion yuan. Owing to its excellent location for transportation, the study area has a total road length of 1231.3 km.

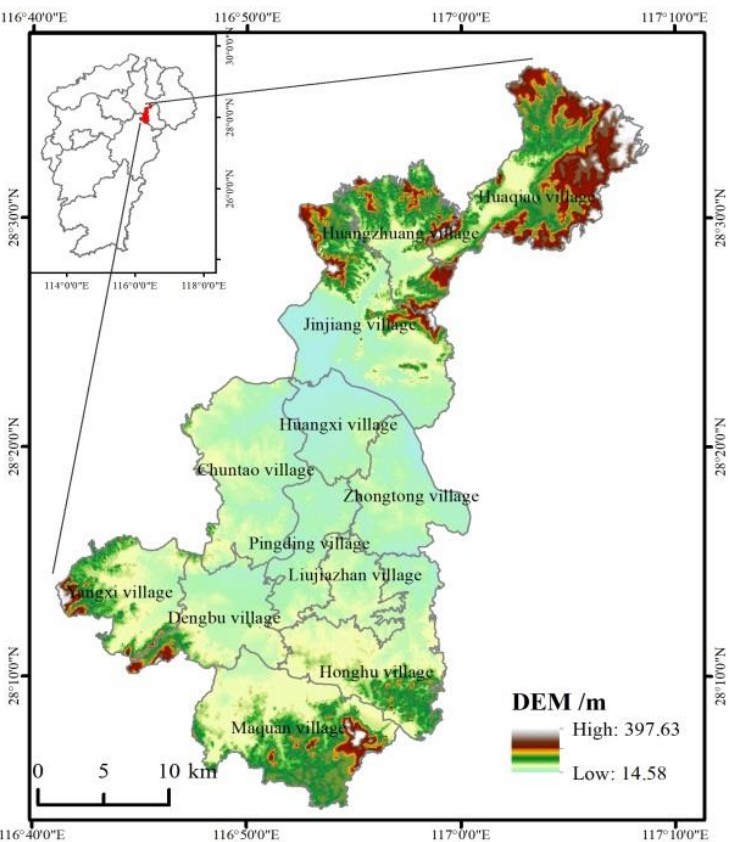

**Figure 1.** Geographical location of the study area—Yujiang District, Yintan City, Jiangxi Province, China.

### 2.2. Data Sources

The 2009, 2013, and 2018 land remote sensing data of Yujiang District were retrieved from a remote sensing monitoring database of China's land use provided by the Resource and Environment Science and Data Center, Chinese Academy of Sciences (https://www.resdc.cn/, accessed on 5 October 2022). We obtained the data by man–machine interactive visual interpretation using ENVI software (Harris Geospatial Solutions Inc., Broomfield, CO, USA) combined with field investigation. The data accuracy was 30 m × 30 m, and the Kappa coefficients of data interpretation in all three periods were >80%, which met the application requirements.

According to the land use classification system of the Resource and Environment Science and Data Center, we classified land use types in the study area using the Extract by Mask tool in ArcGIS v10.2 software (Environment System Research Institute Inc., Redlands, CA, USA) based on the boundary vector map of Yujiang District combined with the actual situation of land use. There were eight land use types: cultivated land, forest land, grassland, urban land, rural residential land, other construction land, water area, and unused land. Digital elevation model (DEM) data were downloaded from the Geospatial Data Cloud (http://www.gscloud.cn, accessed on 5 October 2022) with a spatial resolution of 30 m.

Socio-economic data (total population and gross regional product of 2009–2018) were derived from the Statistical Yearbook of the Yujiang District Statistics Bureau. Meteorological data (average annual rainfall and temperature of 2009–2018) were obtained from the China Meteorological Data Service Center (http://data.cma.cn, accessed on 5 October 2022).

### 2.3. Land Use Change Analysis

Analysis of the dynamic change in land use types is essential for understanding the trend of land use change. We used a transfer matrix to analyze the transfer direction and

quantity of different land use types in Yujiang District across three periods (2009, 2013, 2018). The general form of the land use transfer matrix was selected based on published studies [21,22].

### 2.4. Landscape Metrics and Moving Window Analysis

A range of landscape metrics were used to analyze landscape pattern change in Yujiang District with regard to landscape fragmentation, shape, and diversity [23]. At the class level, we selected the largest patch index (LPI), patch density (PD), mean patch area (MPS), edge density (ED), and landscape shape index (LSI). At the landscape level, we selected the LPI, PD, ED, MPS, LSI, aggregation index (AI), and Shannon evenness index (SHEI). As such, landscape pattern change was analyzed from the perspective of landscape fragmentation (LPI, PD, MPS), patch shape (ED, LSI), and diversity (AI, SHEI) using Fragstats v4.2.1 software (Department of Forest Science, Oregon State University, Corvallis, OR, USA) [24].

Moving window analysis can spatialize and visualize the landscape metrics, for which the choice of window size is crucial. If the window is too small, the local characteristics of the landscape may mask its overall characteristics, leading to insufficient image continuity. If the window is too large, the loss of landscape information may result in fuzzy images [25]. To avoid irrational window size settings that would increase the spatial heterogeneity in landscape patterns, we set different window lengths and generated the corresponding change maps of four landscape metrics (LPI, LSI, AI, SHEI). We then used the Create Fishnet and Extract Multi Values to Points tools of ArcGIS v10.2 to extract landscape metric values from the created Fishnet Label. Further, we used the GS+ software (Gamma Design Software, LLC., Plainwell, USA) to simulate the semi-variograms of landscape metrics under different window lengths and then calculated their nugget–sill ratios to determine the optimal moving window size.

An odd multiple of 30 m was used as the moving window size and a total of 12 window radii were set within the interval of 90–810 m (Figure 2). When the window length ranged between 450 and 630 m, variation in the nugget–sill ratios of LPI, LSI, AI, and SHEI diminished. Accordingly, 570 m was determined to be the optimal window size for moving window analysis. The window moved from the upper left of the study area to analyze the whole study area. The generated landscape metrics map was then analyzed using the Raster Calculator in ArcGIS v10.2 to obtain the spatial change map of landscape metrics for 2009–2013 and 2013–2018.

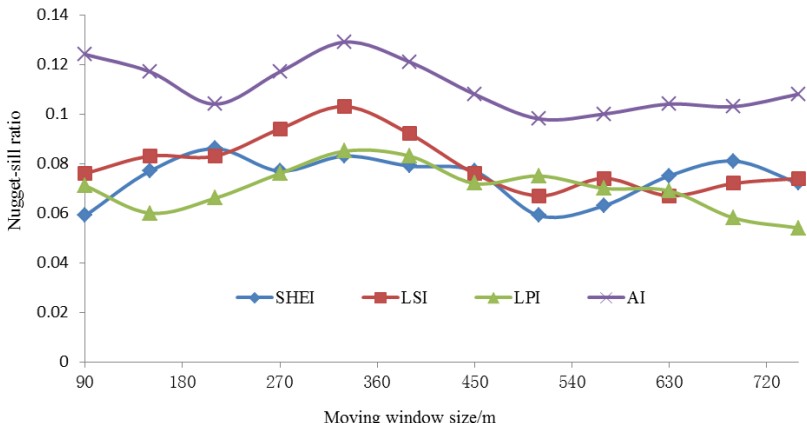

**Figure 2.** Changes in the nugget–sill ratios of landscape metrics with different moving window sizes. SHEI: Shannon evenness index; LSI: landscape shape index; LPI: largest patch index; AI: aggregation index.

### 2.5. Landscape Driving Factor Selection and Geographic Detector Model

The evolution of landscape patterns is a historical synthesis of human activities and natural elements. Considering that Yujiang District is a typical hilly area in southern China, we selected elevation, slope gradient, and slope direction as natural driving factors.

As human activities are important factors influencing the landscape pattern, we selected population density, gross regional product, and distances from major railways, highways, and waterways as socio-economic driving factors. Additionally, meteorological elements cannot be ignored for their impact on the overall landscape pattern, so average annual precipitation and temperature were selected as climate driving factors.

The geodetector developed by Wang et al. [19] can quantitatively describe the stratified heterogeneity of geographical elements, identify the factors driving spatial differentiation in geographical phenomena, and determine the relative contribution of various factors. The model consists of four detector modules: the ecological detector, factor detector, interaction detector, and risk detector. We used the factor detector module to analyze the contribution of various factors driving the change in the overall landscape pattern in Yujiang District. Because this model necessitates the input of a discrete categorical variable as the independent variable $X$, we discretized the continuously changing independent variable factors into 10 categories based on the natural breakpoint method. Then, we determined the explanatory power of each driving factor $X$ in relation to the spatial change of class-level landscape metric $Y$ in 2009–2018 based on the $q$ statistic (Equation (1)). The range of $q$ values is $0 \leq q \leq 1$, and a larger $q$ value is indicative of a stronger explanatory power of factor $X$ [26].

$$q = 1 - \frac{\sum\limits_{h=1}^{L} N_h \sigma_h^2}{N \sigma^2} \tag{1}$$

where $L$ is the category of driving factor $X$; $N_h$ and $N$ are the numbers of units in layer $h$ and in the whole domain, respectively; and $\sigma_h^2$ and $\sigma^2$ are the variances of class-level landscape metric $Y$ for layer $h$ and for the whole domain, respectively.

## 3. Results

### 3.1. Land Use Change

In 2009, the major land use types in Yujiang District were cultivated land (42.89%) and forest land (38.02%). Grassland, water area, rural residential land, and urban land only accounted for 1.19%, 8.14%, 6.21%, and 0.98% of the total study area, respectively. At this stage, the study area was dominated by agriculture and forestry with a low level of urbanization (Figure 3, Table 1). In 2018, the land use showed substantial changes, as indicated by the increased area of urban land (by 33.27%), other construction land (by 21.23%), and rural residential land (by 19.42%), and the decreased area of forest land (by 5.41%), grassland (by 18.18%), and water area (by 2.19%). In the past 10 years, construction land area expanded considerably, whereas forest land, grassland, and water areas all decreased.

From 2009 to 2013, urban land and rural residential land encroached on the surrounding cultivated land at relatively high rates, and the resulting changes were most pronounced in urban areas such as Dengbu Town and Jinjiang Town (Figure 4). In total, the area of forest land, cultivated land, and grassland shrank by 1096.34, 55.83, and 81.34 ha, respectively. From 2013 to 2018, land use change occurred at remarkably higher rates than that observed between 2009–2013. Forest land and grassland showed the largest transfer-out area of 817.55 and 119.66 ha, respectively. In contrast, the area of urban land and rural residential land expanded by 208.6 and 277.72 ha, respectively. Spatially, the conversion of cultivated land and grassland to urban land mainly occurred around the urban area and other central towns, extending in a divergent manner to the northern and southern mountainous areas. The conversion of other land use types mainly occurred in the central flat area, with low intensity but high dispersity.

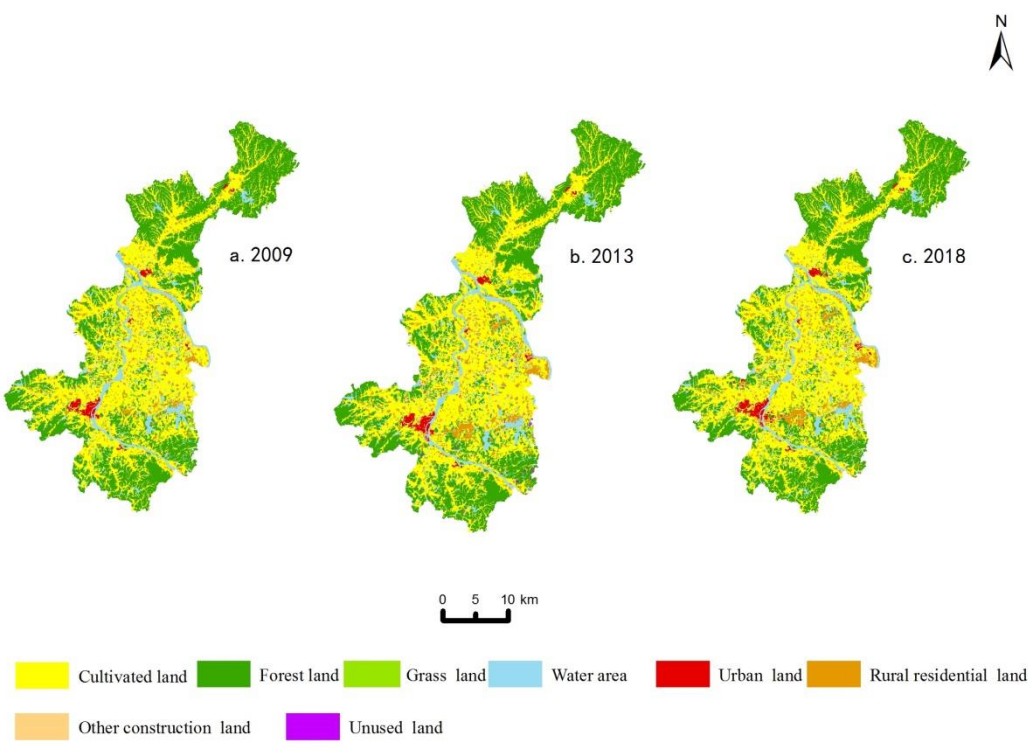

**Figure 3.** Land use change in Yujiang District from 2009 to 2018.

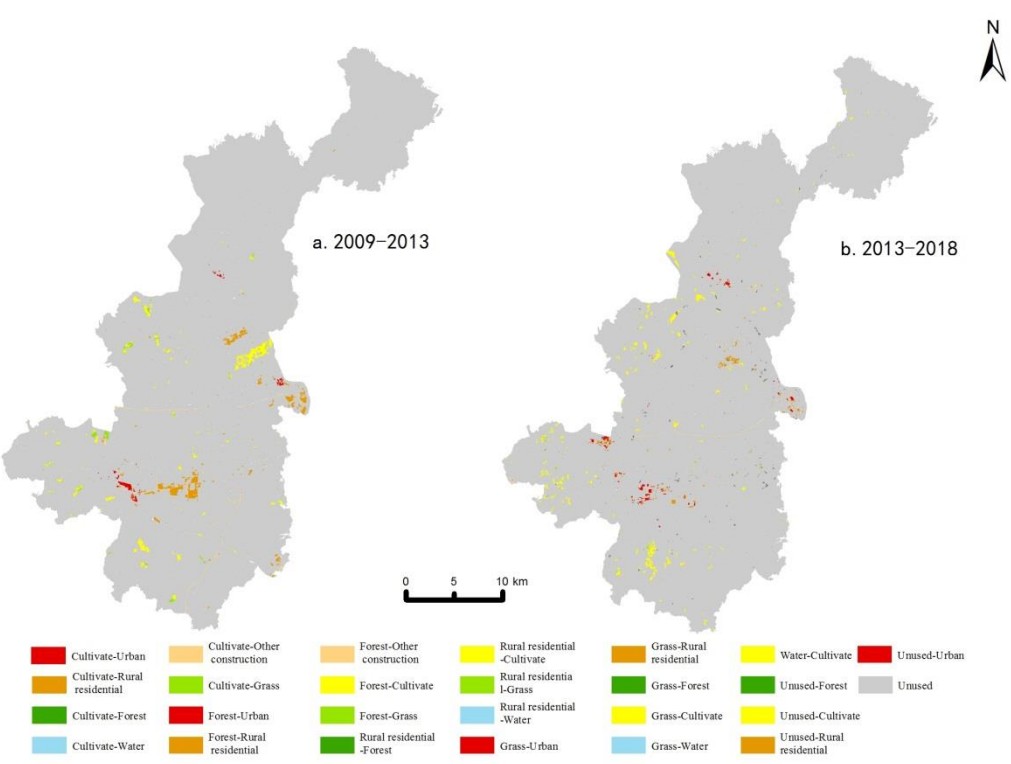

**Figure 4.** Spatial distribution of land use change in Yujiang District from 2009 to 2018.

**Table 1.** Change of land use structure in Yujiang District from 2009 to 2018.

| Year | Variable | Cultivated Land | Forest Land | Grass Land | Water Area | Urban Land | Rural Residential Land | Other Construction Land | Unused Land |
|---|---|---|---|---|---|---|---|---|---|
| 2009 | Area (hm$^2$) | 39,936.34 | 35,403.39 | 1105.71 | 7581.81 | 913.03 | 5778.28 | 2027.72 | 359.73 |
| | Ratio (%) | 42.89 | 38.02 | 1.19 | 8.14 | 0.98 | 6.21 | 2.18 | 0.39 |
| 2013 | Area (hm$^2$) | 39,880.51 | 34,307.05 | 1024.36 | 7562.00 | 1008.20 | 6622.76 | 2337.91 | 363.21 |
| | Ratio (%) | 42.83 | 36.85 | 1.10 | 8.12 | 1.08 | 7.11 | 2.51 | 0.39 |
| 2018 | Area (hm$^2$) | 40,362.47 | 33,489.49 | 904.70 | 7415.85 | 1216.79 | 6900.48 | 2458.19 | 358.03 |
| | Ratio (%) | 43.35 | 35.97 | 0.97 | 7.96 | 1.31 | 7.41 | 2.64 | 0.38 |
| 2009–2013 | Area of change (hm$^2$) | −55.83 | −1096.34 | −81.34 | −19.81 | 95.17 | 844.48 | 310.19 | 3.48 |
| | Change ratio (%) | −0.14 | −3.10 | −7.36 | −0.26 | 10.42 | 14.61 | 15.30 | 0.97 |
| 2013–2018 | Area of change (hm$^2$) | 481.95 | −817.55 | −119.66 | −146.15 | 208.60 | 277.72 | 120.28 | −5.19 |
| | Change ratio (%) | 1.19 | −2.44 | −13.23 | −1.97 | 17.14 | 4.02 | 4.89 | −1.45 |
| 2009–2018 | Area of change (hm$^2$) | 426.13 | −1913.89 | −201.01 | −165.96 | 303.77 | 1122.20 | 430.47 | −1.71 |
| | Change ratio (%) | 1.07 | −5.41 | −18.18 | −2.19 | 33.27 | 19.42 | 21.23 | −0.47 |

*3.2. Landscape Pattern Change*

3.2.1. Landscape Fragmentation

From 2009 to 2018, the LPI and MPS decreased by 7.00% and 12.44%, respectively, whereas PD increased by 14.21% at the landscape level, indicating an increasing trend of landscape fragmentation in the study area. At the class level, the LPI mainly increased in urban land, rural residential land, and other construction land, whereas the opposite trend was observed for grassland, cultivated land, and water area (Figure 5). Among them, the LPI of rural residential land showed the largest increase, from 0.2067 to 0.3927 (89.99%), with a 14.35% increase in the LPI of other construction land. The LPI of grassland decreased the most, from 0.08 to 0.04 (nearly 50%), with a 6.9% decrease in the LPI of cultivated land.

In the past 10 years, PD showed an upward trend across all land use types, with the largest increase recorded for urban land (450.78%), followed by forest land (27.12%), grassland (24.42%), other construction land (22.58), and rural residential land (15.03%; Figure 5). Excluding rural residential land, the MPS of all land use types variably decreased, and the largest decreases were observed for urban land (75.89%), grassland (34.15%), and forest land (25.00%).

From 2009 to 2013, the change in the LPI was concentrated near the urban area, in contrast to sporadic changes in other areas. From 2013 to 2018, the LPI changed dramatically in the central flat area and around traffic trunk lines, with minimal change in the northern and southern areas (Figure 5). Overall, landscape fragmentation increased in the study area over the 10-year period and the center of change was around the urban area and in the central flat area, but a minimal change in landscape fragmentation was noted in the southern and northern mountainous areas with less human disturbance.

3.2.2. Landscape Patch Shape

From 2009 to 2018, both the LSI and ED increased at the landscape level, by 4.31% and 4.83%, respectively, which indicates that the overall landscape patch shape became more complicated (Figure 6). At the class level, the LSI values of different land use types showed variable increases in the past 10 years, among which urban land had the largest increase (50.53%). A smaller increase was observed for the LSI values of other construction land and rural residential land, which increased by 11.45% and 4.82%, respectively. The ED of urban land increased by 73.77%, which is much larger than the increase observed for the ED of other construction land and rural residential land (22.6% and 14.56%, respectively). The results indicate that the change in landscape patch shape in the study area was mainly affected by the change in urban land, other construction land, and rural residential land.

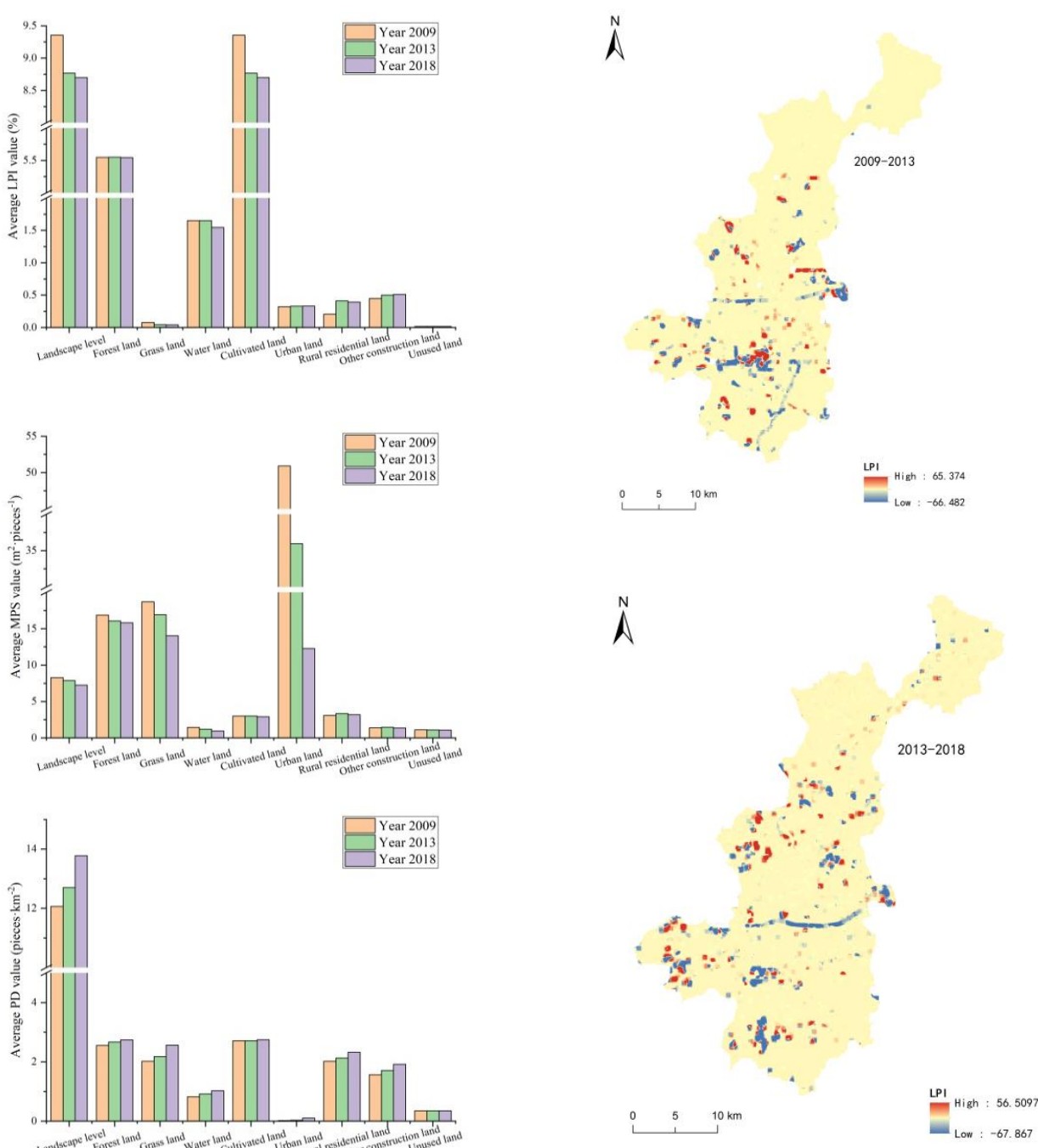

**Figure 5.** Change in landscape fragmentation in Yujiang District from 2009 to 2018.

The spatial changes of the LSI and ED were mainly concentrated in the urban area and around major traffic arteries, which represented the primary area of patch shape change in Yujiang District. However, the LSI exhibited a decreasing trend in the central urban area during 2013–2018 compared with that of the 2009–2013 period (Figure 6), indicating the regularization of landscape shape.

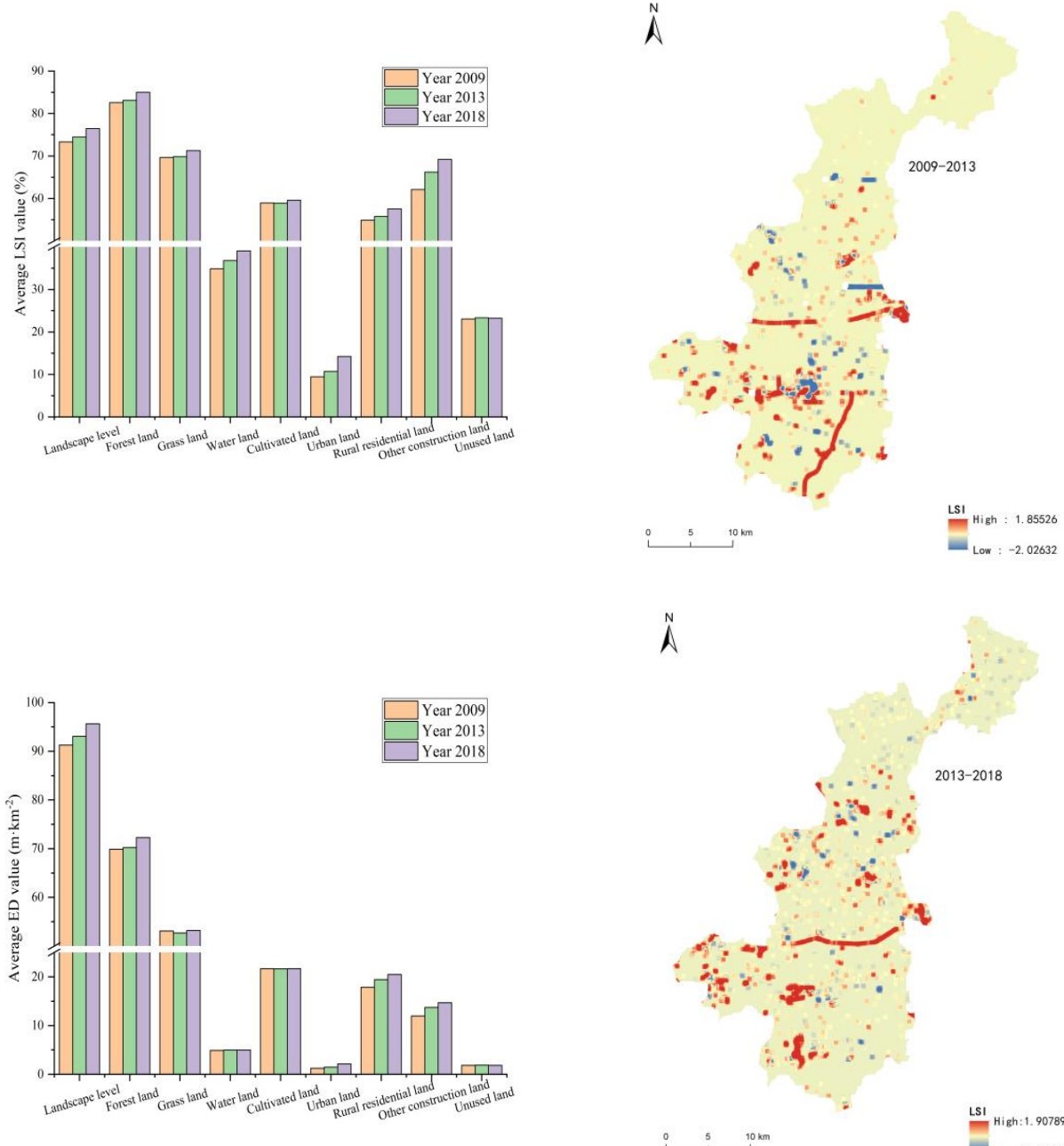

**Figure 6.** Change in landscape patch shape in Yujiang District from 2009 to 2018.

### 3.2.3. Landscape Diversity

From 2009 to 2018, the AI decreased by 0.75% whereas SHEI increased by 2.51% at the landscape level, indicating an increasing landscape diversity in the study area (Figure 7). However, the intensity of the diversity change was lower than that of landscape fragmentation and patch shape change. At the class level, the AI values of most land use types decreased over the past 10 years, except that the AI of rural residential land increased by 1.03%.

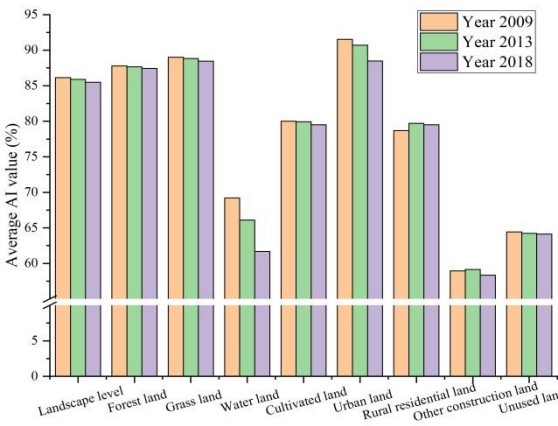

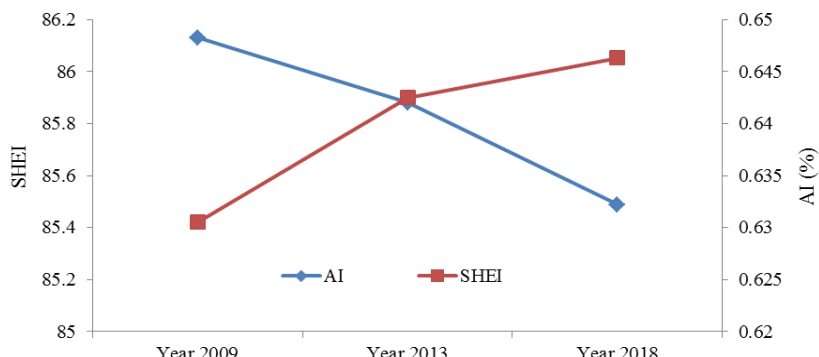

**Figure 7.** Change in landscape diversity in Yujiang District from 2009 to 2018.

### 3.3. Factors Driving Landscape Pattern Change

Using the factor detector module of the geodetector, we evaluated the explanatory power of various factors driving the spatio-temporal evolution of the landscape pattern in the study area. Based on four selected landscape metrics, the explanatory power of 10 factors was ranked in the order of X1 > X4 > X9 > X2 > X10 > X5 > X8 > X3 > X6 > X7 (Figure 8). The factors represented by DEM (X1, X2) which indicates natural surface morphology, as well as gross regional product (X4, X5) and distance from major traffic highways (X9) which reflect economic development status, had strong influence on landscape pattern change. The results indicate that, in addition to natural conditions, human socio-economic activities posed a profound impact on landscape pattern change. Among the natural factors, average annual temperature and rainfall had a limited ability to explain landscape pattern change.

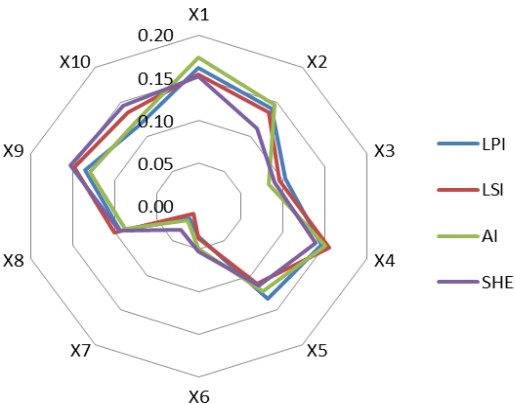

**Figure 8.** The *q*-value radar map of factors driving landscape pattern evolution in Yujiang District. X1: elevation; X2: slope gradient; X3: slope direction; X4: gross regional product; X5: population density; X6: average annual temperature; X7: average annual rainfall; X8: Distance from major railways; X9: Distance from major highways; X10: Distance from major waterways. LPI: largest patch index; LSI: landscape shape index; AI: aggregation index; SHEI: Shannon evenness index.

## 4. Discussion

In the past decade, the landscape pattern of Yujiang District has changed prominently, with a strong dependence on local topography and geomorphology. The overall terrain of the study area is high in the north and south, and gradually becomes gentle toward the central part. Among its diverse landforms, hills followed by mountains are predominant. As a typical hilly area in southern China, it has an average elevation of 100–300 m. Therefore, it is unsurprising that elevation (X1) and slope gradient (X2) represented by DEM are the key natural factors contributing to the spatio-temporal changes of landscape pattern, which is consistent with the results of previous studies [27,28]. To ensure a rational layout of landscape classes such as production, life, and ecology, key constraints that drive landscape pattern evolution must be taken into account and the scientific layout should be adjusted based on local conditions [29].

The reform policy of the rural homestead system is another key factor influencing landscape pattern change in the study area [30]. In 2015, Yujiang District was listed as one of the first pilot counties to reform the rural homestead system. Before the reform, local farmers arbitrarily built houses with no planning, keeping houses in close proximity to neighboring houses. There were serious problems such as households having more than one house sites and building new houses without demolishing the old ones. All these phenomena led to disorderly expansion of villages. In recent years, Yujiang District has vigorously promoted the reform of the rural homestead system to ensure one homestead for one household and unified planning. The considerably larger increase in LPI (89.99%) relative to that in PD (15.03%), together with the minor increase in MPS (3.64%) and AI (1.03%) from 2009 to 2018, indicates aggregation of rural residential land in the study area. This demonstrates that the rural homestead system reform policy is an essential factor contributing to landscape pattern optimization in rural residential land [31]. Our finding is consistent with the conclusion of Fu et al. who investigated the spatio-temporal changes in rural residential land in Yujiang District before and after the rural homestead system reform [32]. Further, Yujiang District should continue to promote a new round of rural homestead system reform and keep optimizing the landscape pattern of rural residential land.

Appropriate selection of driving factors and analysis models is the key to characterizing landscape pattern evolution in a given area [33]. To unravel the mechanisms that drive landscape pattern change in Yujiang District, we quantitatively explain the influence of various factors on landscape pattern evolution using the factor detector module of geodetector. It should be noted that our study still needs to be further improved. First, a range of natural factors can be selected as independent variables, and the explanatory power of the

selected variables may be enhanced by analyzing their influence under interactions with unnatural factors. Additionally, statistics and machine learning can be integrated to build boosted regression models for correlation analysis between various influencing factors [34]. For example, based on a boosted regression tree model, Wu et al. determined the major factors influencing the wetland landscape pattern in the Shuangyang River Basin [35].

From the perspective of landscape ecology, the combination of scale effect analysis with spatial landscape pattern metrics can prevent the loss of information caused by landscape heterogeneity [36]. In this study, we have fully considered the relationship between landscape pattern characteristics and scale, and set a total of 12 different moving window sizes to determine the optimal scale for the study area. This provides effective methodological support for a thorough analysis of the spatio-temporal evolution of the landscape pattern in typical low-mountain and hilly areas in southern China. It should be noted that there are certain differences in the responses of various landscape pattern metrics to scale effects, and some subjective factors may interfere with the identification of the change interval of inflection points [37]. Therefore, future studies should pay continuous attention to theoretical and methodological exploration of the framework for landscape scale–process–pattern analysis. It is also necessary to improve the accuracy of spatio-temporal evolution analysis of landscape patterns by a combination of geography- and ecology-related models and methods.

## 5. Conclusions

This study quantified and visualized the processes of land use and landscape pattern change in a typical hilly area in southern China from 2009 to 2018 using ArcGIS v10.2 and Fragstats v4.2.1 software. The optimal moving window size suitable for the study area was determine using semi-variogram analysis conducted with GS+ software, which could eliminate the effect of scale on the original landscape pattern change. Further, the key factors driving landscape pattern change were identified by using geodetector.

Over the past 10 years, the dominant landscape classes (cultivated land, forest land) remained unchanged in Yujiang District, whereas other landscape classes changed distinctly. The expansion of urban land, other construction land, and rural residential land was mainly concentrated in the urban area and central towns as well as near the major traffic lines. In contrast, the area of forest land, grassland, and water area shrank over time.

Using an odd multiple of 30 m as the moving window size, a total of 12 window radii were set within the interval of 90–810 m. The variation in the nudge–sill ratios of landscape metrics diminished with window lengths of 450–630 m. Accordingly, 570 m was selected as the optimal window size for the analysis of landscape pattern evolution.

At the landscape level, the study area had a more complex patch shape over time, with increased landscape fragmentation and diversity. At the class level, cultivated land, forest land, and grassland all showed landscape fragmentation, whereas urban land exhibited scattered expansion. The aggregation of rural residential land was closely related to vigorous promotion of rural homestead system reform and enhanced intensification of rural land use.

The land use structure and landscape pattern change in the study area were mainly driven by topographic, socio-economic, and traffic factors. Natural factors such as average annual temperature and rainfall had minimal influence on the overall landscape pattern.

**Author Contributions:** Conceptualization, J.Z.; methodology, J.Z., Y.Z. and X.Z.; software, J.Z.; validation, J.Z., J.G. and X.H.; formal analysis, J.Z. and M.L.; investigation, X.Z.; resources, X.Z.; data curation, J.Z. and X.Z.; writing—original draft preparation, J.Z.; writing—review and editing, J.Z. and J.G.; supervision, X.Z.; project administration, X.Z.; funding acquisition, X.Z. All authors have read and agreed to the published version of the manuscript.

**Funding:** This research was funded by the Chinese national key project of high-resolution Earth observation system, grant number 82-Y50G22-9001-22/23.

**Data Availability Statement:** All data in this study can be found in the research data source in the paper. The land remote sensing data of 2009, 2013 and 2018 were obtained from the Data Center for Resources and Environmental Sciences, Chinese Academy of Sciences (https://www.resdc.cn/, accessed on 5 October 2022); DEM data were obtained from the geospatial data cloud (http://www.gscloud.cn, accessed on 5 October 2022); socio-economic data were obtained from the Yujiang District Statistical Yearbook. Meteorological data are from the China Meteorological Science Data Sharing Network (http://data.cma.cn, accessed on 5 October 2022).

**Acknowledgments:** The authors would like to thank the anonymous reviewers for their valuable comments and remarks.

**Conflicts of Interest:** The authors declare no conflict of interest.

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
