# Peer review of "Spatio-Temporal Evolution and Driving Factors of Landscape Pattern in a Typical Hilly Area in Southern China: A Case Study of Yujiang District, Jiangxi Province"

_forests, doi:10.3390/f14030609_

Round 1

Reviewer 1 Report

Congratulations for this Manuscript. It is so interesting for spatio-temporal evolution of landscape in Southern China. The MS is well written, and every section was well organized.

1. Please, include the geodetector in the abstract to explain about the model.

2. Line 96, remove "here" word.

L98. Please, mention the figure 1 before the location. It is not mentioned in the main text.

L137. Please, add a reference about the software.

L286-339. Please, add more references in this section. there are just two references in the current version of MS. I kindly recommend to add more references and improve this section.

Author Response

Point 1: Please, include the geodetector in the abstract to explain about the model.

Response 1: As suggested by the reviewer, I have listed using the geographic detector model to analyze the spatio-temporal changes of landscape pattern in a typical hilly area (Yujiang District, Yingtan City, Jiangxi Province) in southern China in the abstract(Line12-15).

Point 2: Line 96, remove "here" word.

Response 2: I have deleted “here” word(Line99).

Point 3:L98. Please, mention the figure 1 before the location. It is not mentioned in the main text.

Response 3: As suggested by the reviewer, I have noted “Figure 1” where appropriate in main text (Line97).

Point 4: L137. Please, add a reference about the software.

Response 4: I have added the Fragstats v4.2.1 software reference (Line140-141).

Point 5: L286-339. Please, add more references in this section. there are just two references in the current version of MS. I kindly recommend to add more references and improve this section.

Response 5: As suggested by the reviewer, I have improved the discussion section and added references (Line 351-407).

We would like to thank the referee again for taking the time to review our manuscript. Look forward to hearing from you.

Reviewer 2 Report

Dear Authors,

Thank you for this very clear study. Interesting and well structurate.

I suggest to improve the conclusion. In the text you find the note.

Good luck  

Author Response

The conclusion refers to a summary of you present in the discussion. I suggest to emphasize the methodologies used in the study. Explain the novelties of this approach also considering the literature mentioned in the introduction

Response : We thank the reviewer for pointing this out. The methods used in this study are highlighted in the conclusion of the paper.(lines 408–434).

We would like to thank the referee again for taking the time to review our manuscript. Look forward to hearing from you.

Reviewer 3 Report

Review

Manuscript ID: Forests - 2236175

“Spatio-temporal Evolution and Driving Factors of Landscape Pattern in a Typical Hilly Area in Southern China: A Case Study of Yujiang District, Jiangxi Province”

The landscape pattern is an arrangement of different types and quantities of landscape elements in spatial structure and location. It is also a specific combination of natural and socio-economic factors over complex spatio-temporal scales. Basing on the multi-period remote sensing data, authors of the manuscript selected optimal scale (570 m), to analyze the spatio-temporal changes of landscape pattern in a typical hilly area of Yujiang District, located in southern China. The reviewed manuscript brings forward an important issues for characterizing the specifis regions, and the research problem is presented in an interesting way. The manuscript is 14-pages long, with 27 references to literature, and I would recommend to add some more adequate references by expanding the discussion section, which seems not to be sufficiently developed.
Throughout the text there have been noted minor grammatical errors, that should be corrected. Please, check again the whole text, in order to remove the above-mentioned shortcomings, e.g.:

-  Line 23: provide suport (results - plural);

- Line 29: of-landscape?

- Line 344: it should be „.. study area was determined by using ..”

Author Response

Throughout the text there have been noted minor grammatical errors, that should be corrected. Please, check again the whole text, in order to remove the above-mentioned shortcomings.

Response: We are very sorry for these grammatical errors. We have checked the whole text and made improvements in the revised manuscript.

We would like to thank the referee again for taking the time to review our manuscript. Look forward to hearing from you.